# Double-Needle Yamane Technique Using Flanged Haptics in Ocular Trauma—A Retrospective Survey of Visual Outcomes and Safety

**DOI:** 10.3390/jcm10122562

**Published:** 2021-06-09

**Authors:** Katarzyna Nowomiejska, Dariusz Haszcz, Maksymilian Onyszkiewicz, Tomasz Choragiewicz, Aleksandra Czarnek-Chudzik, Agata Szpringer-Wabicz, Katarzyna Baltaziak, Agnieszka Brzozowska, Mario Damiano Toro, Robert Rejdak

**Affiliations:** 1Chair and Department of General and Pediatric Ophthalmology, Medical University of Lublin, 20-059 Lublin, Poland; haszcz@umlub.pl (D.H.); maksymilian.onyszkiewicz@gmail.com (M.O.); tomekchor@wp.pl (T.C.); olaczarnek@poczta.pl (A.C.-C.); agatasara@gmail.com (A.S.-W.); k.baltaziak@gmail.com (K.B.); toro.mario@email.it (M.D.T.); robert.rejdak@umlub.pl (R.R.); 2Department of Mathematics and Medical Biostatistics, Medical University of Lublin, 20-059 Lublin, Poland; agnieszka.brzozowska@umlub.pl; 3Faculty of Medicine, Collegium Medicum, Cardinal Stefan Wyszyński University, 01-815 Warsaw, Poland

**Keywords:** vitrectomy, intraocular lens (IOL), posterior segment, anterior segment surgery, intraocular lens implantation (IOL implantation), intraocular lens surgery (IOL surgery), ocular trauma surgery, yamane technique

## Abstract

To evaluate visual outcomes and safety of the double-needle technique using flanged haptics (Yamane technique) in patients with aphakia caused by ocular trauma at a trauma referral center. Retrospective: Consecutive interventional case series of 30 patients who underwent the Yamane technique due to posttraumatic aphakia. The double-needle technique using flanged haptics was combined with anterior vitrectomy (group A) in 14 patients, and with pars plana vitrectomy (PPV) (group B) due to retinal detachment, nucleus dislocation into the vitreous cavity, or intraocular lens (IOL) displacement in 16 patients. No intraoperative complications were noted. There was significant improvement in the visual acuity in both groups at the second postoperative visit. However, the visual acuity was significantly worse in the group treated with the Yamane technique combined with PPV. Silicone oil tamponade in PPV group was associated with worse visual acuity, whereas post lensectomy status was associated with poor visual function result in the anterior vitrectomy group. There was one case of slight IOL decentration and one retinal detachment during the postoperative follow-up period in the group with PPV. In this case series, the Yamane technique applied in traumatized eyes was found to be an efficacious and safe procedure. Combining the Yamane technique with PPV due to posterior segment ocular trauma was associated with worse functional results in the follow-up at three months. Further studies with longer follow-up evaluations are required to verify long-term complications.

## 1. Introduction

Ocular injury is one of the major issues in the public health care system, still being one of the significant causes of blindness worldwide. The epidemiology of ocular trauma depends on the age, part of the world, sport activity or socioeconomic status of the patient [1,2,3]. The classification of the mechanical eye injury terminology which received recognition from many international ophthalmologist societies was defined by Kuhn and colleagues in 1996 [4].

As a result of ocular trauma, the anatomical conditions of the eye can dramatically change and result in inadequate support for placement of intraocular lens (IOL) in the capsular bag [5]. However, when there is no capsular or iris support, IOLs can be fixated in eyes using both sutured [6,7,8,9,10,11] or sutureless methods [12,13,14,15,16,17]. Yamane [16] recently described a new mini-invasive technique of scleral fixation with two needles, which externalize the haptics of a three-piece IOL. A flanged haptic tip is created and cauterized to be fixed intrasclerally. It has already been described in postoperative aphakia related to complicated cataract surgery and dislocated IOL [18].

To the best of our knowledge, the efficacy and safety of the double-needle technique using flanged haptics in ocular trauma patients has not been investigated to date.

The aim of this study was to describe the functional and anatomical results of ocular trauma aphakia cases managed with the secondary implantation of the IOL using the Yamane technique.

## 2. Materials and Methods

This is a retrospective interventional surgical case series of consecutive posttraumatic aphakic patients at the Chair and Department of General and Pediatric Ophthalmology of the Medical University of Lublin, Poland, which is the tertiary center for eye injury in the south-eastern Poland. The study includes the data of surgeries performed in a time period from May 2019 to February 2021. This study followed the tenets of the Declaration of Helsinki. Approval of the Ethics Committee at the Medical University of Lublin, Poland was given (n° KE-0254/88/2020). The patients gave their written informed consent.

The study population included 30 consecutive patients diagnosed with posttraumatic aphakia. The inclusion criteria were as follows: aphakia and history of ocular blunt trauma within 6 months prior to the onset of surgery. Patients were excluded in the case of active ocular inflammation, infection or glaucoma. The data extracted from medical records included: pre- and postoperative best-corrected visual acuity (BCVA), applanation tonometry, slit-lamp evaluation, and fundus examination. BCVA was determined using Snellen charts and converted into logMAR (minimum angle of resolution). Subjects with counting fingers, hand motion, light perception, or no light perception visual acuity were assigned with values of logMAR of 1.9, 2.3, 2.7, and 3.0, respectively [19,20]. Intra- and postoperative complications were collected to evaluate the safety of the procedure. Functional success was represented by changes in BCVA from the baseline to the most recent follow-up.

All patients had routine preoperative biometric measurement with IOL Master (Carl Zeiss Meditec, Jena, Germany). Having awareness that refractive outcomes of the Yamane technique are less predictable than standard cataract surgery we calculated all IOL powers by SRK-T formula using the IOL Master (Carl Zeiss Meditec, Jena, Germany) as for the bag power. Visual acuity, intraocular pressure, and IOL position were assessed after 2 weeks, 6 weeks, and 3 months postoperatively or more often if indicated. The primary anatomic outcomes were the optic pupillary centration and location of haptic externalization. Postoperatively, a steroid (Dexafree^®^, dexamethasone sodium phosphate 1% solution eye drops, Laboratoires Thea, France) and an antibiotic (Oftaquix^®^, 0.5% levofloxacin ophthalmic solution, Santen Oy, Finland) were given 5 times per day for 2 weeks and then only the steroid was administered 3 times per day during the following 2 weeks.

### 2.1. Surgical Technique

The surgeries were done in general anesthesia in 8 cases or peribulbar anesthesia in 22 cases. IOL was implemented using the Yamane technique (Yamane et al., 2014) by one surgeon (DH) in all cases. Anterior vitrectomy (group A) or 23 G pars plana vitrectomy (PPV) (Forth Worth, Alcon) (group B) was performed before IOL fixation depending on the status of the retina and vitreous. 

A 30-gauge two separate needles were used to perform angled sclerotomies placed 2 mm from the limbus and located 180° from each other. A three-piece IOL (Alcon MA60AC) was injected through the limbal incision into the anterior chamber. To prevent the IOL from falling into the vitreous cavity, the trailing haptic was kept outside. Next, the leading haptic was inserted into the needle’s lumen. The same maneuver was done again 180° opposite to fixate the second trailing haptic. Both haptics have been and externalized onto the conjunctiva through the transscleral tunnel and the IOL position was centered with simultaneous double-needle withdrawal. With low temperature cautery the externalized haptic end was heated to create a triangular flange with a diameter of 0.3 mm, which was pushed back and fixed intrasclerally (Appendix A).

### 2.2. Statistical Analysis

The values of all analyzed parameters were presented using the mean value, median, quartiles, and standard deviation. The normality of the distribution of the analyzed parameters was evaluated using the W Shapiro–Wilk test. Mann–Whitney test was used to compare independent groups. In the statistical analysis, data-mining was also used. A *p* value ≤ 0.05 was considered statistically significant. All calculations were performed using STATISTICA 13.0 (StatSoft, Krakow, Poland) software.

## 3. Results

Most patients were males (22 patients). The mean patient age was 52 years (range: from 22 to 79 years) and the mean follow-up was 3 months (range: from 2 to 4 months). Fourteen patients underwent anterior vitrectomy (Group A, Table 1) and 16 patients had additional PPV (Group B, Table 2) due to coexisting retinal detachment (8), dropped nucleus (4), post PPV+ silicone oil-silicone oil removal (3), and IOL displacement (1). In 11 PPV cases, 5000 Cst silicone oil was used as a tamponade, while air was used in 4 cases and BSS in 1 case. All patients underwent uneventful surgery.

In Group A, BCVA improved in 12 cases, remained the same in one case (post PPV + subluxated cataract), and worsened in 1 case with subluxated cataract. In group B, BCVA improved in 13 cases and remained the same in 3 cases with retinal detachment. The preoperative and postoperative visual outcomes in Group A and B are summarized in Table 1 and Table 2, respectively. Additionally, no intraoperative complications were observed in both groups. Postoperatively, one decentration with retinal detachment was observed, which were treated with another PPV with silicone oil as tamponade. 

There were no statistically significant differences in visual acuity between the anterior vitrectomy and PPV groups preoperatively. During the observation period at the first follow-up visit, BCVA was significantly worse in the PPV group compared with the anterior vitrectomy group (*p* = 0.03). Moreover, significant differences between the groups were also found at the last postoperative visit (*p* = 0.002), and the results were worse in the PPV group as well (median 1.30 vs. 0.25). Statistical analysis showed significant improvement in visual acuity after both PPV (*p* = 0.00001) and anterior vitrectomy (*p* = 0.0007) (Table 3).

In both groups the greatest improvement in BCVA was observed between preoperative period and the last visit (Table 4).

Data mining analysis in regard to factors predicting good visual acuity showed that silicone oil tamponade in PPV group was associated with worse visual acuity, whereas post lensectomy status was associated with poor visual function result in the anterior vitrectomy group.

Preoperatively, intraocular pressure (IOP) was significantly higher in the group with anterior vitrectomy (*p* = 0.003), postoperatively, there were no significant differences in the values of IOP between both groups (*p* > 0.05) (Table 5).

## 4. Discussion

Ocular trauma surgery is a worldwide growing branch of ophthalmology. Accumulating evidence suggests that a huge progress in understanding the pathophysiology of vision loss linked with eye injury has been made during the recent years [21,22,23]. Furthermore, the development of new instrumentations and techniques of IOL implantation or vitreoretinal surgery in injured eyes has improved the efficacy and safety of ocular trauma management [24,25,26]. The Yamane technique has gained popularity worldwide, and several modifications have been described [27,28]. It has now become preferred approach for IOL fixation when there is no adequate capsular support for a traditional IOL. However, studies published so far include heterogeneous groups of patients with complicated past ocular histories and co-morbidities including trauma, retinal detachment, and corneal disease [27,29], which add additional variability to the visual acuity. To the best of our knowledge this is the first study focused only on posttraumatic aphakia.

In our study, a significant visual improvement was achieved in surgeries performed with both anterior vitrectomy and PPV, and all surgeries were completed uneventfully. However, after 3 months of the follow-up, visual acuity was significantly worse in the PPV group; it is caused by the presence of the silicone oil in these eyes and coexisting damage of the posterior segment which declines the prognosis in regard to visual function. As complications, one eye developed a retinal detachment and IOL decentration.

In a previous study of 100 eyes, Yamane and coworkers [30] who introduced this technique, observed as a complication an iris capture by the IOL in 8 eyes (8%), vitreous hemorrhage in 5 eyes (5%), and cystoid macular edema in 1 eye (1%). However, no trauma cases were described.

In another study, Kelkar [29] and colleagues described a modified Yamane technique with a 27G needle used in 31 consecutive patients with aphakia due to complicated cataract surgery, lensectomy, and dislocated IOL. There were no intraoperative complications, and two decentrations were reported postoperatively.

To date, there are few studies on the efficacy and safety of the Yamane technique. However, it seems to be a quite an easy and fast procedure in the hands of an experienced surgeon, with a short learning curve. It requires no special instruments, and no special IOL has to be ordered in advance. It is a minimally invasive method of IOL intrascleral fixation. Some modifications of this technique have been described, e.g., the as trailing-haptic-first technique [31], the handshake riveting flanged technique [32] and the flanged intrascleral intraocular lens fixation with a single needle [33].

Nowadays, posttraumatic aphakia may be supplied using both transscleral or iris fixation of IOL. These two methods have been compared by Saleh and coworkers in a group of 23 eyes with traumatic cataract and severely damaged capsular bags [34]. Group 1 consisted of 8 eyes which underwent intrascleral IOLs [Acrysof MN60 AC], and group 2 comprised retropupillary iris-claw IOLs (Verisyse). The final VA did not differ between groups. No intraoperative complications occurred in group 2; one haptic broke in group 1 and macular edema occurred in both groups. We did not observe any macular oedema in our series of 30 patients who underwent the Yamane technique.

In a large retrospective analysis of secondary IOL implantation, Vounotrypidis [35] in a large retrospective analysis of secondary IOL implantation observed that previous vitrectomy seems to be a risk factor for a worse BCVA in eyes that undergo secondary IOL implantation. Also, in our study, BCVA was worse in eyes that had undergone PPV before IOL implantation, whereas eyes (11/14) that underwent only anterior vitrectomy achieved BCVA > 0.4.

IOL fixated using the Yamane technique may need repositioning Recently, Pugazhendhi [36] has described the first double-needle Yamane repositioning in the eye of a patient with pigmentary dispersion syndrome who previously underwent Yamane-fixated IOL. The lens was rubbing against the iris, which caused pigment dispersion and elevated intraocular pressure (with all topical antihypertensive medications) despite the presence of iridotomy. The surgeon used the double-needle technique to fix the new IOL. The surgeon used double-needle technique to fix new IOL. The reposition was successful, the visual acuity improved (from 20/200 to 20/40) and IOP was reduced to normal limits. In our study none of the eyes required repositioning.

Mustafi and colleagues [37] described outcomes of 45 eyes with aphakia, including ocular trauma. The eyes were managed with sutureless secondary intraocular lens fixation with haptic flanging, but no PPV was done in studied eyes.

Yamane technique circumvents the need for sutures, glue, and scleral or conjunc-tival dissection, thus it is very sophisticated method of secondary IOL implantation. It is particularly important in the case of traumatized eyes with tissue defects and destruction.

The shortcoming of our study is that no anterior segment imaging device (optical coherence tomography) for measuring the optic tilt was used and the corneal endothelial cell density was not assessed. Ian and colleagues [38] evaluated IOL tilt in 17 patients using SS-OCT and the IOL hepatic symmetry in scleral tunnel. There was no significant difference between the horizontal and vertical tilt, and there were no statistically significant differences between the distance between the center of the haptic flange and the scleral spur of the nasal and temporal haptic. Additionally, since angle recession may be one of the effects of ocular trauma, the double-needle technique (as sutureless and scleral fixation) might be considered as an option of IOL implantations to avoid high levels of intraocular pressure. Moreover, it is a retrospective study and the samples’ sizes in are quite small at 14 and 16 subjects. However, due to relative rarity of posttraumatic aphakia and diversity of ocular trauma cases, it is difficult to obtain large groups of patients and make prospective study. Most of the studies in this field are only case series [37,39]. In our study, the follow-up period was only 3 months, thus a longer term follow-up (including after silicone oil removal) in the future study would be helpful to determine long term visual prognosis with this technique, as well as changes in IOP or development of glaucoma. The strength of this study is that all the cases were performed by the same surgeon and all the cases used the same lens.

## 5. Conclusions

The double-needle technique can be used in secondary IOL implantation in posttraumatic aphakic eyes with good results and a low complication rate. In comparison to iris fixation, the secondary IOL implantation technique decreases the probability of an increase in intraocular pressure to negligible. Additionally, it seems to have a quite short learning curve; thus, it can be used not only by highly skilled surgeons. However, further randomized clinical studies with a long-term follow-up period and a wider number of patients are necessary to demonstrate the safety and effectiveness of the procedure in aphakic patients.

## Figures and Tables

**Table 1 jcm-10-02562-t001:** Preoperative and postoperative visual outcomes in eyes with posttraumatic aphakia treated with the Yamane technique followed by anterior vitrectomy (Group A).

N°	Diagnosis	Best Corrected Visual Acuity (LogMAR)		
		Preoperatively	First Visit Postoperatively	Last Visit Postoperatively
1	IOL displacement	0.7	0.7	0.4
2	Subluxated cataract	1.9	0.4	0.15
3	Aphakia	2.7	0.7	0.4
4	Subluxated cataract	0.3	0.3	0.15
5	Post PPV + lensectomy	0.3	0.7	0.15
6	Subluxated cataract	0.1	1.00	0.3
7	Post lensectomy	1.9	0.5	0.3
8	Subluxated cataract	1.9	1.9	1.9
9	Post lensectomy	1.9	0.3	0.15
10	Post PPV + subluxated cataract	1.00	1.00	1.00
11	Subluxated cataract	1.9	0.5	0.5
12	IOL displacement	2.3	0.7	0.2
13	Aphakia	1.9	0.15	0.15
14	Post PPV + subluxated cataract	2.3	0.046	0.046

N°: number of patients; PPV: pars plana vitrectomy; IOL: intraocular lens.

**Table 2 jcm-10-02562-t002:** Preoperative and postoperative visual outcomes in eyes with posttraumatic aphakia treated with the Yamane technique followed by pars plana vitrectomy (PPV) (Group B).

N°	Diagnosis	Post PPV Tamponade	Best Corrected Visual Acuity (LogMAR)		
			Preoperatively	First Visit Postoperatively	Last Visit Postoperatively
1	Dropped nucleus	Air	0.5	0.2	0.2
2	Retinal detachment	Silicone oil	2.7	1.3	1.3
3	Post PPV + oil	Silicone oil removal + air	1.9	1.3	1.3
4	Retinal detachment	Silicone oil	1.9	0.15	0.15
5	Retinal detachment, post keratoplasty	Silicone oil	2.3	2.3	IOL Decentration, Retinal detachment; 2.3
6	Retinal detachment, hemorrhage	Silicone oil	2.3	2.0	1.8
7	Retinal detachment post cerclage, and displacement of IOL	Silicone oil	2.3	2.3	2.3
8	Retinal detachment, hemorrhage	Silicone oil	2.7	1.8	1.8
9	Dropped nucleus	BSS	0.7	0.4	0.4
10	Post PPV + silicone oil	Silicone oil removal + air	1.00	1.00	0.5
11	Retinal detachment	Silicone oil	2.0	1.9	1.8
12	Post PPV + silicone oil	Silicone oil	1.9	1.8	1.8
13	Dropped nucleus	Silicone oil	1.00	0.7	0.7
14	Aphakia	Silicone oil	2.3	2.3	2.3
15	IOL displacement	Silicone oil removal + air	1.9	0.5	0.5
16	Dropped nucleus	Silicone oil	1.9	0.7	0.7

N°: number of patients; PPV: pars plana vitrectomy; IOL: intraocular lens; BSS-balanced salt solution.

**Table 3 jcm-10-02562-t003:** Assessment of the visual acuity (logMAR) in both studied groups (anterior vitrectomy –AV and pars plana vitrectomy-PPV).

Vitrectomy	Mean	Standard Deviation	Lower Quartile	Median	Upper Quartile	Statistical Significance
Preoperatively						
PPV	1.83	0.68	1.45	1.90	2.30	0.23
AV	1.51	0.85	0.70	1.90	1.90	
First visit postoperatively						
PPV	1.29	0.78	0.60	1.30	1.95	0.03 *
AV	0.64	0.46	0.30	0.60	0.70	
Last visit postoperatively						
PPV	1.24	0.79	0.50	1.30	1.80	0.002 *
AV	0.41	0.49	0.15	0.25	0.40	

* Statistically significant.

**Table 4 jcm-10-02562-t004:** Assessment of the visual acuity (logMAR) between the studied periods.

Visual Acuity Improvement between Preoperative and Last Visit Period	Mean	Standard Deviation	Lower Quartile	Median	Upper Quartile
PPV ^^^	0.59	0.57	0.15	0.40	1.05
Anterior vitrectomy ^#^	1.09	0.96	0.15	1.50	1.75

^ Z = 2.69, *p* = 0.007*; ^#^ Z = 1.88, *p* = 0.06*.

**Table 5 jcm-10-02562-t005:** Changes in intraocular pressure before and after operation.

Vitrectomy	Mean	Standard Deviation	Lower Quartile	Median	Upper Quartile	Statistical Significance	
						Z	*p*
Preoperatively							
PPV	15.88	6.14	15.00	16.00	16.00	−2.97	0.003 *
AV	19.57	5.24	17.00	18.00	21.00		
First visit postoperatively							
PPV	13.88	5.95	9.00	16.00	17.00	−1.85	0.06
AV	18.21	4.58	16.00	17.00	18.00		
Last visit postoperatively							
PPV	13.06	7.08	5.50	15.50	18.00	−0.85	0.86
AV	15.86	3.78	13.00	16.00	17.00		

* Statistically significant.

## Data Availability

The data presented in this study are available on request from the corresponding author.

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
