# Peer review of "Double-Needle Yamane Technique Using Flanged Haptics in Ocular Trauma—A Retrospective Survey of Visual Outcomes and Safety"

_jcm, 2021, doi:10.3390/jcm10122562_

Round 1

Reviewer 1 Report

This is overall a well written paper. There are some limitations to the retrospective design of the study, since the visual potential and an atomic changes of the eyes that requires pars plans vitrectomy after trauma may be different than those that underwent anterior vitrectomy. Therefore, it is difficult to make conclusions from comparisons between the two groups. Nevertheless, the study has important descriptive value for real-world application of this important surgical technique. Longer term follow-up (including after silicone oil removal) would be helpful to determine long term visual prognosis with this technique, as well as changes in IOP or development of glaucoma.

Author Response

Reviewer 1

  • This is overall a well written paper. There are some limitations to the retrospective design of the study, since the visual potential and an anatomic changes of the eyes that requires pars plans vitrectomy after trauma may be different than those that underwent anterior vitrectomy. Therefore, it is difficult to make conclusions from comparisons between the two groups. Nevertheless, the study has important descriptive value for real-world application of this important surgical technique.

In the discussion chapter (line 273) it has been added:

Moreover, it is retrospective study and the samples’ sizes in are quite small - 14 and 16 subjects. However, due to relative rarity of posttraumatic aphakia and diversity of ocular trauma cases, it is difficult to obtain large groups of patients and make prospective study. Most of the studies in this field are only case series (mustafi, mayer).

Additionally, two new references have been added:

Mustafi D, Tom E, Messenger WB, Banitt MR, Rezaei KA. Outcomes of sutureless secondary intraocular lens fixation with haptic flanging in a cohort of surgically complex eyes. Graefes Arch Clin Exp Ophthalmol. 2021 May;259(5):1357-1363.

Mayer C, Baur ID, Storr J, Khoramnia R. Complete anterior segment reconstruction: Corneal transplantation and implantation of an iris prosthesis and IOL in a single surgery. Eur J Ophthalmol. 2021 Jan 28:1120672121991052.

  • Longer term follow-up (including after silicone oil removal) would be helpful to determine long term visual prognosis with this technique, as well as changes in IOP or development of glaucoma.

In the discussion chapter it has been added (line 276):

In our study follow-up period was only 3 months, thus longer term follow-up (including after silicone oil removal) in the future study would be helpful to determine long term visual prognosis with this technique, as well as changes in IOP or development of glaucoma.

Reviewer 2 Report

  1. using a scleral lamellar dissection (this is a modification of Yamane- why the need for lamellar dissection?
  2. Where are the haptics externalised? Dissections at 2 mm, is lens placed at 2 mm or further back? How di they calculate lens power? (2.5mm suggests in bag power)
  3. 'Burn as you go' technique- makes the second haptic enclavation very difficult due to angles- did they modify needle (double bend) ddi they get instances of haptic kink?
  4. Group A had anerior vitrectomy but two cases also PPV? This does not make sense
  5. discussion suggests both anterior and Posterior Vity did best- this makes no sense as a posterior Vitrectomy will include removing anterior vitreous

Author Response

Reviewer 2

  • Using a scleral lamellar dissection (this is a modification of Yamane- why the need for lamellar dissection?

Thank you for this important remark. We have analysed again the videos recorded during surgeries and we have modified the section describing in details the surgical method as follows (line 100):

A 30-gauge two separate needles were used to perform angled sclerotomies placed 2 mm from the limbus and located 180° from each other. A three-piece IOL (Alcon MA60AC) was injected through the limbal incision into the anterior chamber. To prevent the IOL from falling into the vitreous cavity, the trailing haptic was kept outside. Next, the leading haptic was inserted into the needle’s lumen. The same maneuver was done again 180° opposite to fixate the second trailing haptic. Both haptics have been and externalized onto the conjunctiva through the transscleral tunnel and the IOL position was centered with simultaneous double-needle withdrawal. With low tem-perature cautery the externalized haptic end was heated to create a triangular flange with a diameter of 0.3 mm, which was pushed back and fixed intrasclerally (video).

Additionally, the video with surgical technique has been attached as supplemental material.

  • Where are the haptics externalised? Dissections at 2 mm, is lens placed at 2 mm or further back?

In the methods section it has been written (line 105):

Both haptics have been and externalized onto the conjunctiva through the transscleral tunnel and the IOL position was centered with simultaneous double-needle withdrawal. With low temperature cautery the externalized haptic end was heated to create a triangular flange with a diameter of 0.3 mm, which was pushed back and fixed intrasclerally  (video).

Additionally, the video with surgical technique has been attached as supplemental material.

  • How did they calculate lens power? (2.5mm suggests in bag power)

In the methods section (line 81) it has been added: Having awareness that refractive outcomes of the Yamane technique are less predictable than standard cataract surgery we calculated all IOL powers by SRK-T formula using the IOL Master (Carl Zeiss Meditec, Jena, Germany) as for the bag power.

  • 'Burn as you go' technique- makes the second haptic enclavation very difficult due to angles- did they modify needle (double bend) ddi they get instances of haptic kink?

We have not observed difficulties during second haptic enclavation. The needle has not been modified and we didn’t get instances of haptic kink.

  • Group A had anterior vitrectomy but two cases also PPV? This does not make sense

In the abstract it is written (line 20): The double-needle technique using flanged haptics was combined with anterior vitrectomy (group A) - in 14 patients and with pars plana vitrectomy (PPV) (group B) due to retinal detachment, nucleus dislocation into the vitreous cavity, or intraocular lens (IOL) displacement - in 16 patients.

In the methods section (line 94) it is written:

Anterior vitrectomy (group A) or pars plana vitrectomy (PPV) (group B) was performed before IOL fixation depending on the status of the retina and vitreous.

And later (line 120) in the results section: Fourteen patients underwent anterior vitrectomy (Group A, Table 1) and 16 patients had additional PPV (Group B, Table 2) due to coexisting retinal detachment (8), dropped nucleus (4), post PPV+ silicone oil-silicone oil removal (3), and IOL displacement (1).

  • discussion suggests both anterior and Posterior Vity did best- this makes no sense as a posterior Vitrectomy will include removing anterior vitreous.

In the discussion chapter it is now written (line 214): In our study, a significant visual improvement was achieved in surgeries per-formed with both anterior vitrectomy and PPV and all surgeries were completed une-ventfully. However, after 3 months of the follow-up visual acuity was significantly worse in PPV group. It is caused by the presence of the silicone oil in these eyes and coexisting damage of the posterior segment which declines the prognosis in regard to visual function.

Reviewer 3 Report

The manuscript is well - written, however it lacks novelty.  Another issue is that the sample size is too small (n=14 Group A & n=16 Group B) . 

Author Response

Reviewer 3

  • The manuscript is well - written, however it lacks novelty.

In the discussion chapter it has been added (line 206):

The Yamane technique has gained popularity worldwide, and several modifications have been described (Jacob, randerson). It has now become preferred approach for IOL fixation when there is not adequate capsular support for a traditional IOL. However, studies published so far included a heterogeneous groups of patients with complicated past ocular histories and co-morbidities including trauma, retinal detachment, and corneal disease (randerson, kelkar) which adds additional variability to the visual acu-ity. To the best of our knowledge this is the first study focused only on posttraumatic aphakia.

Additionally, two new  references have been added:

Randerson EL, Bogaard JD, Koenig LR, Hwang ES, Warren CC, Koenig SB. Clinical Outcomes and Lens Constant Optimization of the Zeiss CT Lucia 602 Lens Using a Modified Yamane Technique. Clin Ophthalmol. 2020;14:3903-3912. Published 2020 Nov 17.

Jacob S, Kumar DA, Rao NK. Scleral fixation of intraocular lenses. Curr Opin Ophthalmol. 2020;31(1):50–60.

  • Another issue is that the sample size is too small (n=14 Group A & n=16 Group B) .

In the discussion chapter (line 273) it has been added:

Moreover, it is retrospective study and the samples’ sizes in are quite small - 14 and 16 subjects. However, due to relative rarity of posttraumatic aphakia and diversity of oc-ular trauma cases, it is difficult to obtain large groups of patients and make prospective study. Most of the studies in this field are only case series (mustafi, mayer).

Additionally, two new references have been added:

Mustafi D, Tom E, Messenger WB, Banitt MR, Rezaei KA. Outcomes of sutureless secondary intraocular lens fixation with haptic flanging in a cohort of surgically complex eyes. Graefes Arch Clin Exp Ophthalmol. 2021 May;259(5):1357-1363.

Mayer C, Baur ID, Storr J, Khoramnia R. Complete anterior segment reconstruction: Corneal transplantation and implantation of an iris prosthesis and IOL in a single surgery. Eur J Ophthalmol. 2021 Jan 28:1120672121991052.

Round 2

Reviewer 3 Report

Despite the revisions made and the authors/ clarifications, I still believe that the originality/ novelty is low and the sample size is too small.